# Screener: Learning Conditional Distribution of Dense Self-supervised Representations for Unsupervised Pathology Segmentation in 3D Medical Images

## Abstract

Accurate and automated anomaly segmentation is critical for assisting clinicians in detecting and diagnosing pathological conditions, particularly in large-scale medical imaging datasets where manual annotation is not only time- and resource-intensive but also prone to inconsistency. To address these challenges, we propose SCREENER, a fully self-supervised framework for visual anomaly segmentation, leveraging self-supervised representation learning to eliminate the need for manual labels. Additionally, we model the conditional distribution of local image patterns given their global context, enabling the identification of anomalies as patterns with low conditional probabilities and assigning them high anomaly scores.

SCREENER comprises three components: a descriptor model that encodes local image patterns into self-supervised representations invariant to local-content-preserving augmentations; a condition model that captures global contextual information through invariance to image masking; and a density model that estimates the conditional density of descriptors given their global contexts to compute anomaly scores.

We validate SCREENER by training a fully self-supervised model on over 30,000 3D CT images and evaluating its performance on four large-scale test datasets comprising 1,820 3D CT scans across four chest and abdominal pathologies. Our framework consistently outperforms existing unsupervised anomaly segmentation methods. Code and pre-trained models will be made publicly available.

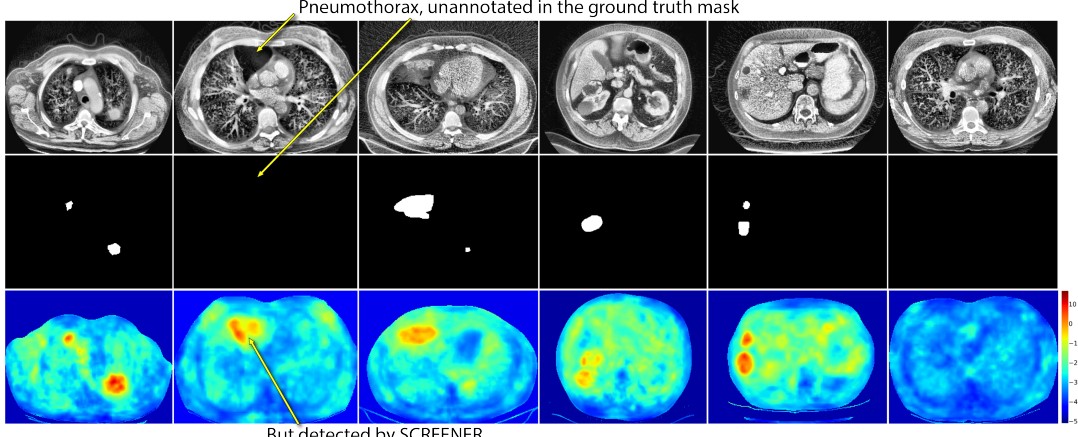

Figure 1: Examples of 2D slices of 3D medical CT images (the first row), the ground truth masks of their pathological regions (the second row) and the anomaly maps predicted by fully self-supervised SCREENER for pathology segmentation (the third row). Note that, the second image from the left contains pneumothorax, missed by ground truth annotation mask, but detected by SCREENER.

# 1 INTRODUCTION

The accurate and automated segmentation of pathologies in medical computed tomography (CT) images is crucial for assisting clinicians in diagnosing and treating various conditions. However, developing supervised models for pathology segmentation faces significant challenges: labeled datasets are scarce, annotations often cover only a limited range of findings, and manual labelling is not only resource-intensive but also inconsistent. For example, in Figure 1, pneumothorax is present in the second column (black region framed by red box) but is not included in the ground truth mask. Hence, supervised methods for pathology segmentation are often constrained in scope and applicability.

In contrast, large-scale datasets of unlabelled CT images are readily available through public repositories (Team, 2011; Ji et al., 2022; Qu et al., 2024). These datasets remain largely underutilized due to the lack of annotations, despite their potential to enable fully unsupervised learning approaches. Leveraging this abundance of unlabelled data, we aim to develop a model capable of distinguishing pathological regions from normal ones without requiring labeled training data. Our core assumption is that pathological patterns are significantly rarer than healthy patterns in random CT images. This motivates framing pathology segmentation as an unsupervised visual anomaly segmentation (UVAS) problem, where anomalies correspond to pathological regions.

While existing UVAS methods have been explored extensively for natural images, their adaptation to medical imaging remains challenging. A major hurdle is that most CT datasets contain unannotated pathological regions, and there is no automatic way to filter these out to ensure a training set composed entirely of normal (healthy, non-pathological) images — a common requirement for synthetic-based (Zavrtanik et al., 2021; Marimont & Tarroni, 2023) and reconstruction-based (Baur et al., 2021; Schlegl et al., 2019) UVAS methods.

Density-based approaches (Gudovskiy et al., 2022; Zhou et al., 2024), which assume anomalies are rare rather than entirely absent, are better suited for this setting, as they can handle training datasets with unannotated pathological regions. These methods model normal patterns probabilistically and assign higher anomaly scores to deviations. However, they rely on encoders pre-trained on ImageNet (Deng et al., 2009), optimized for natural images and not for the unique structures and textures in medical CT images. This domain shift leads to suboptimal feature representations failing to capture subtle pathological variations, reducing their effectiveness in medical settings.

To address these challenges, we propose SCREENER, a framework that enhances density-based UVAS through domain-specific self-supervised learning and learned contextual conditioning. To avoid domain shift issues and labelling requirement, we pre-train self-supervised encoders (O Pinheiro et al., 2020; Wang et al., 2021; Bardes et al., 2022; Goncharov et al., 2023) to produce dense CT-specific feature maps. We further introduce a second self-supervised encoder that generates masking-invariant representations, capturing global context without being influenced by local anomalies. Finally, we train a conditional density model to predict the feature maps of one encoder based on the outputs of the other. Anomaly scores are assigned to image regions with high prediction errors, enabling effective segmentation of pathological regions.

We demonstrate the effectiveness of SCREENER by training it on over 30,000 3D CT volumes spanning chest and abdominal regions and evaluating its performance on four large-scale test datasets comprising 1,820 scans with diverse pathologies. As shown in Figure 1, our model successfully segments pathological regions across different organs and conditions. We summarize the key contributions of this work:

- **Self-Supervised Representations for UVAS**: We demonstrate that dense self-supervised representations outperform supervised feature extractors in visual anomaly segmentation, enabling a fully self-supervised framework applicable in domains with limited labeled data.

- **Learned Conditioning Variables**: We introduce self-supervised condition variables for density-based models, simplifying the estimation of conditional distributions and achieving remarkable segmentation performance using a simple Gaussian density model.

- **First Large-Scale Study of UVAS in 3D CT Images**: This work presents the first large-scale evaluation of UVAS methods for 3D CT images, showing state-of-the-art performance on unsupervised semantic segmentation of pathologies in diverse anatomical regions, including lung cancer, pneumonia, liver and kidney tumors.

## 2 BACKGROUND & NOTATION

**Density-based UVAS** methods assign high anomaly scores to image regions with rare patterns using two models, which we call a *descriptor model* and a *density model*. The descriptor model encodes image patterns into vector representations, while the density model learns their distribution and assigns anomaly scores based on the learned density.

In existing methods (Gudovskiy et al., 2022; Zhou et al., 2024), the descriptor model $f_{\theta^{\mathrm{desc}}}$ is a fully-convolutional neural network pre-trained on ImageNet. For a 3D image $\mathbf{x} \in \mathbb{R}^{H \times W \times S}$, it produces feature maps $\mathbf{y} \in \mathbb{R}^{h \times w \times s \times d^{\mathrm{desc}}}$, where each position $p \in P$ corresponds to a descriptor $\mathbf{y}[p] \in \mathbb{R}^{d^{\mathrm{desc}}}$. Here, position set $P = \{p \mid p \in [1, \ldots, h] \times [1, \ldots, w] \times [1, \ldots, s]\}$.

The density model $q_{\theta^{\mathrm{dens}}}(y)$ estimates the marginal density $q_Y(y)$ of descriptors. For an abnormal pattern at position $p$, the descriptor $\mathbf{y}[p]$ is expected to lie in a low-density region, yielding a low $q_{\theta^{\mathrm{dens}}}(\mathbf{y}[p])$. Conversely, normal patterns produce high densities. During inference, the negative log-density values, $-\log q_{\theta^{\mathrm{dens}}}(\mathbf{y}[p])$ are used as anomaly segmentation scores. Density models we use in SCREENER are simple Gaussian model and more expressive normalizing flow (see Appendix E).

This framework can be extended using a conditioning mechanism. For each position $p$, an auxiliary variable $\mathbf{c}[p]$, referred to as a *condition*, is introduced. Let $C$ denote the condition at a random position in a random image. Instead of modelling the complex marginal density $q_Y(y)$, the conditional density $q_{Y|C}(y|c)$ is learned for each condition $c$. During inference, the negative log-conditional densities, $-\log q_{\theta^{\mathrm{dens}}}(\mathbf{y}[p] \mid \mathbf{c}[p])$, are used as anomaly scores. State-of-the-art methods (Gudovskiy et al., 2022; Zhou et al., 2024) adopt this conditional framework and use sinusoidal positional encodings as conditions. See detailed descriptions for positional condition alternatives in Appendix D.

**Self-supervised learning** leverages unlabelled data to learn representations invariant to transformations through auxiliary tasks. SSL objectives align embeddings of augmented views $x^{(1)}, x^{(2)}$ of the same image $x$ while avoiding trivial solutions (mapping all images to the same vector). In vision domain, augmentations typically include color jitter and random crops. Representations are derived by feeding inputs $x$ to an encoder $f_\theta$ (a neural network), yielding $z = f_\theta(x)$. We employ adaptations of SimCLR (Chen et al., 2020) and VICReg (Bardes et al., 2021) to dense feature learning (O Pinheiro et al., 2020; Wang et al., 2021; Bardes et al., 2022; Goncharov et al., 2023) in our approach. For detailed description of these methods, please refer to Appendix C.

## 3 METHOD

Here we present our method for unsupervised semantic segmentation of pathological regions in 3D medical CT images, illustrated in Figure 2. Our method introduces two key innovations to the density-based UVAS framework: *self-supervised descriptor model* (Section 3.1), and *self-supervised condition model* (Section 3.2). Section 3.3 describes the training pipeline for density modelling.

### 3.1 DESCRIPTOR MODEL

The descriptor model $f_{\theta^{\mathrm{desc}}}$ is critical to our method. It must produce descriptors $\mathbf{y}[p]$ that distinguish between pathological and normal positions $p$, as this differentiation directly determines the anomaly scores in the density-based UVAS framework. Simultaneously, descriptors should minimize irrelevant information; for instance, if they capture noise from CT images, the density model may assign high anomaly scores to healthy regions with extreme noise, leading to false positives.

To pre-train dense descriptors, we use dense joint embedding SSL methods (Section 2 and Appendix C), which allow explicit control over the information content of the representations. Specifically, we penalize descriptors for failing to distinguish between different positions within or across images, ensuring they capture spatially discriminative features. Simultaneously, we enforce invariance to low-level perturbations, such as cropping and color jitter, to eliminate irrelevant information.

The descriptor model training pipeline is illustrated in the upper part of Figure 2. From a random image $\mathbf{x}$, we extract two overlapping 3D crops of random size, resize them to $H \times W \times S$, and apply random augmentations, such as color jitter. The augmented crops, denoted as $\mathbf{x}^{(1)}$ and $\mathbf{x}^{(2)}$, are fed into the descriptor model, producing feature maps $\mathbf{y}^{(1)}$ and $\mathbf{y}^{(2)}$.

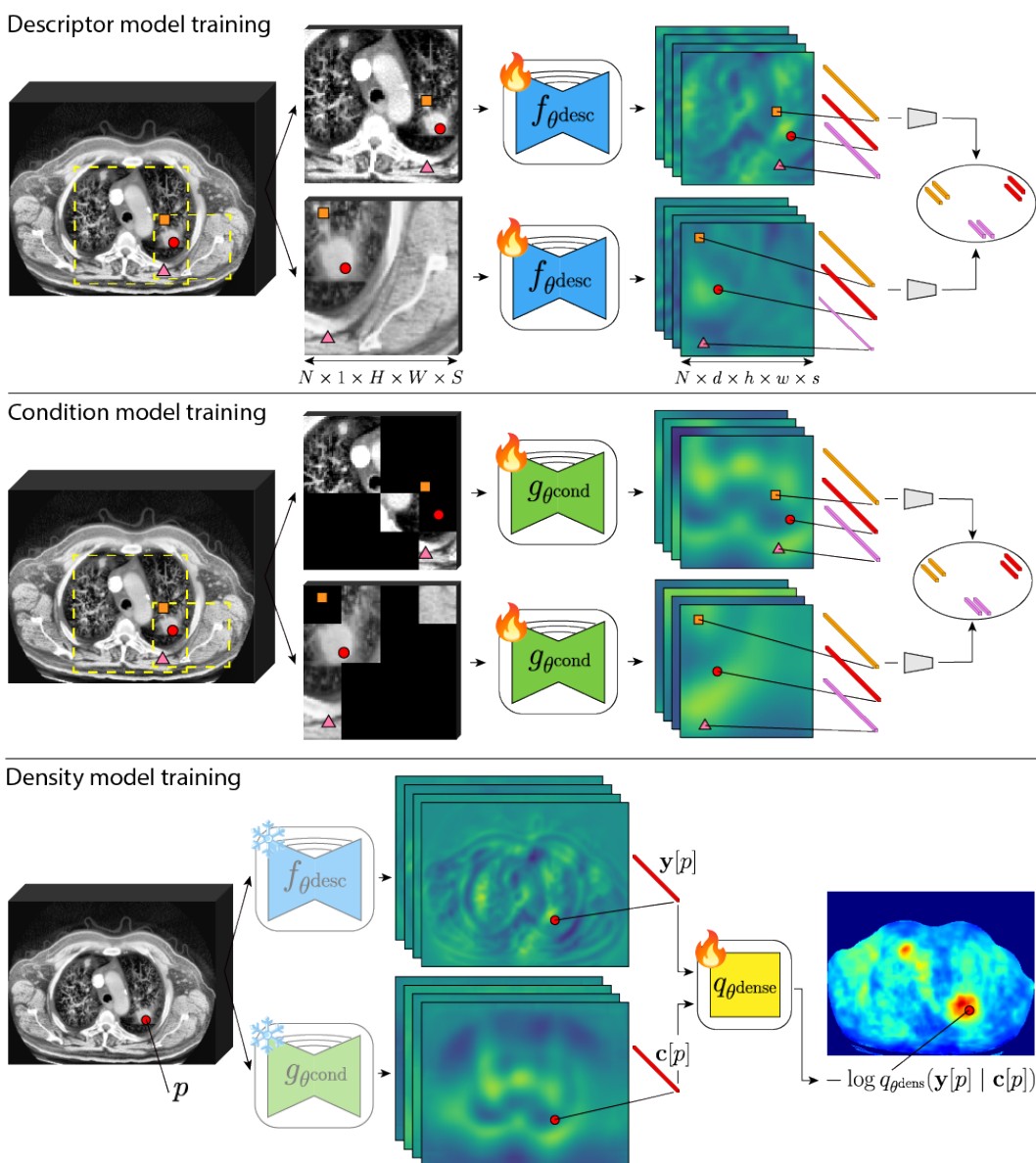

Figure 2: Illustration of SCREENER. First, we train a self-supervised descriptor model to produce informative feature maps invariant to image crops and color jitter. Second, we train a self-supervised condition model similarly but also enforce invariance to random block masking, ensuring its feature maps are insensitive to anomalies and reflect only contextually inferable information. Finally, the density model learns the conditional distribution $p_{Y|C}(y \mid c)$ of feature vectors $Y = y[p]$ and $C = c[p]$ from the descriptor and condition models at a given position $p$. Anomaly score maps are obtained by applying the density model pixel-wise, efficiently implemented by $1 \times 1 \times 1$ convolutions.

From the overlapping region of the two crops, we randomly select $n$ positions. For each position $p$, we compute its coordinates $p^{(1)}$ and $p^{(2)}$ relative to the augmented crops and extract descriptors $y^{(1)} = \mathbf{y}^{(1)}[p^{(1)}]$ and $y^{(2)} = \mathbf{y}^{(2)}[p^{(2)}]$. These descriptors form a *positive pair*, as they correspond to the same position in the original image but are predicted from different augmentations.

Repeating this process for $m$ seed images yields a batch of $N = n \cdot m$ positive pairs, denoted as $\{(y_i^{(1)}, y_i^{(2)})\}_{i=1}^{N}$. This strategy for sampling dense positive pairs follows the approach in (Gon-

charov et al., 2023). Using this batch, we optimize the descriptor model with SSL objectives. In this work, we employ two prominent objectives: InfoNCE (Chen et al., 2020) and VICReg (Bardes et al., 2021), detailed in Appendix C.

## 3.2 CONDITION MODEL

Our self-supervised condition model is inspired by a thought experiment: suppose a region of a CT image is masked, and we attempt to infer its content based on the visible context (as shown in the upper masked crop in Figure 2). In most cases, we would assume the masked region is healthy unless there is explicit evidence to suggest otherwise. This assumption reflects a model of the conditional distribution over possible inpaintings given the context. If the actual content significantly deviates from this expectation –indicating low conditional probability– it is classified as an anomaly.

Building on this intuition, we propose that the condition $\mathbf{c}[p]$ in the conditional density-based UVAS framework should capture the *global* context of the image position $p$. *Global* implies that $\mathbf{c}[p]$ must be inferable from various masked views of the image. At the same time, conditions may vary across different regions of the image to encode position-specific information, such as anatomical location or tissue type.

To achieve these properties, we learn conditions $\mathbf{c}[p]$ through a self-supervised condition model $g_{\theta^{\text{cond}}}$, which has a fully-convolutional architecture similar to the descriptor model. The model generates feature maps $\mathbf{c} \in \mathbb{R}^{h \times w \times s \times d^{\text{desc}}}$ that are invariant to image masking, providing a condition for each position in the input image. The training process mirrors that of the VICReg descriptor model (Section 3.1), with the addition of masking as part of the augmentations. An illustration of this approach is shown in the middle part of Figure 2.

The learned conditions $\mathbf{c}[p]$ are designed to ignore the presence of pathologies, as such information cannot be consistently inferred from masked views. Instead, the condition model likely encodes patient-level attributes (e.g., age, gender) and position-specific attributes (e.g., anatomical region, tissue type) that are predictable from masked contexts. Conditioning on these variables simplifies density estimation, as conditional distributions are often less complex than marginal distributions.

Moreover, conditioning can improve fairness: for instance, if certain anatomical regions or demographic groups are underrepresented in the training data, an unconditional density model might treat these as anomalies. In contrast, a model conditioned on gender or anatomical region would handle such cases more appropriately by treating them within their specific context.

## 3.3 DENSITY MODELS

To train a conditional density model, $q_{\theta^{\text{dense}}}(y \mid c)$, we sample a batch of $m$ random crops, $\{\mathbf{x}_i\}_{i=1}^m$, each of size $H \times W \times S$, from different CT images. Each crop is passed through the pre-trained descriptor and condition models to produce descriptor maps, $\{\mathbf{y}_i\}_{i=1}^m$, and condition maps, $\{\mathbf{c}_i\}_{i=1}^m$, both of size $h \times w \times s$. We then optimize the conditional negative log-likelihood loss:

$$\min_{\theta_{\text{dense}}} \quad \frac{1}{m \cdot |P|} \sum_{i=1}^m \sum_{p \in P} - \log q_{\theta^{\text{dense}}}(\mathbf{y}_i[p] \mid \mathbf{c}_i[p]).$$

At inference, an input CT image is divided into $M$ overlapping patches, $\{\mathbf{x}_i\}_{i=1}^M$, each of size $H \times W \times S$. For each patch, we apply the descriptor, condition, and conditional density models to compute the anomaly map, $\{- \log q_{\theta^{\text{dense}}}(\mathbf{y}_i[p] \mid \mathbf{c}_i[p])\}_{p \in P}$. These patch-wise anomaly maps are upsampled to $H \times W \times S$ and aggregated into a single anomaly map for the entire CT image by averaging predictions in overlapping regions.

We explore two parameterizations for the marginal and conditional density models: Gaussian distributions as a straightforward baseline and normalizing flows as an expressive generative model enabling tractable density estimation. For further details, please refer to Appendix E.

Table 1: Summary information on the datasets that we use for training and testing of all models.

| Dataset | # 3D images | Annotated pathology | # 3D images w/ non-zero pathology mask |
|---|---|---|---|
| NLST (Team, 2011) | 25,652 | – | – |
| AMOS (Ji et al., 2022) | 2,123 | – | – |
| AbdomenAtlas (Qu et al., 2024) | 4,607 | – | – |
| LIDC (Armato III et al., 2011) | 1017 | lung cancer | 603 |
| MIDRC (Tsai et al., 2020) | 115 | pneumonia | 115 |
| KiTS (Heller et al., 2020) | 298 | kidney tumors | 298 |
| LiTS (Bilic et al., 2023) | 117 | liver tumors | 107 |

## 4 EXPERIMENTS

### 4.1 DATASETS

We train all models on three CT datasets: NLST (Team, 2011), AMOS (Ji et al., 2022) and AbdomenAtlas (Qu et al., 2024). Note that we do not use any image annotations during training. Some of the datasets employed additional criteria for patients to be included in the study, i.e. age, smoking history, etc. Note that such large scale training datasets include diverse set of patients, implying presence of various pathologies.

We test all models on four datasets: LIDC (Armato III et al., 2011), MIDRC-RICORD-1a (Tsai et al., 2020), KiTS (Heller et al., 2020) and LiTS (Bilic et al., 2023). Annotations of these datasets include segmentation masks of certain pathologies. Any other pathologies that can be present in these datasets are not labeled. We summarize dataset statistics and pathology information in Table 1.

### 4.2 EVALUATION METRICS

We use standard quality metrics for assessment of visual anomaly segmentation models which are employed in MVTecAD benchmark (Bergmann et al., 2021): pixel-level AUROC and AUPRO calculated up to $0.3$ FPR. We also compute area under the whole pixel-level ROC-curve. Despite, our model can be viewed as semantic segmentation model, we do not report standard segmentation metrics, e.g. Dice score , due to the following reasons. As we mention in Section 4.1, available testing CT datasets contain annotations of only specific types of tumors, while other pathologies may be present in the images but not included in the ground truth masks. It makes impossible to fairly estimate metrics like Dice score or Hausdorff distance, which count our model's true positive predictions of the unannotated pathologies (see second image from the left in the Figure 1 for example) as false positive errors and strictly penalize for them. However, the used pixel-level metrics are not sensitive to this issue, since they are based on sensitivity and specificity. We estimate sensitivity on pixels belonging to the annotated pathologies. To estimate specificity we use random pixels that do not belong to the annotated tumors which are mostly normal, thus yielding a practical estimate.

Table 2: Quantitative comparison of our best model and the existing unsupervised visual anomaly segmentation methods on pathology segmentation in 3D medical CT images.

| Model | AUROC | | | | AUROC up to FPR0.3 | | | | AUPRO up to FPR0.3 | | | |
|---|---|---|---|---|---|---|---|---|---|---|---|---|
| | LIDC | MIDRC | KiTS | LiTS | LIDC | MIDRC | KiTS | LiTS | LIDC | MIDRC | KiTS | LiTS |
| Autoencoder | 0.71 | 0.65 | 0.66 | 0.68 | 0.31 | 0.21 | 0.24 | 0.25 | 0.59 | 0.24 | 0.26 | 0.37 |
| f-AnoGAN | 0.82 | 0.66 | 0.67 | 0.67 | 0.52 | 0.21 | 0.24 | 0.22 | 0.46 | 0.18 | 0.24 | 0.22 |
| DRAEM | 0.63 | 0.72 | 0.82 | 0.83 | 0.21 | 0.31 | 0.50 | 0.51 | 0.17 | 0.20 | 0.50 | 0.57 |
| MOOD-Top1 | 0.79 | 0.79 | 0.77 | 0.80 | 0.43 | 0.43 | 0.40 | 0.46 | 0.32 | 0.29 | 0.40 | 0.32 |
| MSFlow | 0.70 | 0.66 | 0.64 | 0.64 | 0.26 | 0.20 | 0.18 | 0.17 | 0.21 | 0.14 | 0.19 | 0.17 |
| Screener (ours) | **0.96** | **0.89** | **0.90** | **0.94** | **0.89** | **0.68** | **0.69** | **0.80** | **0.66** | **0.46** | **0.68** | **0.66** |

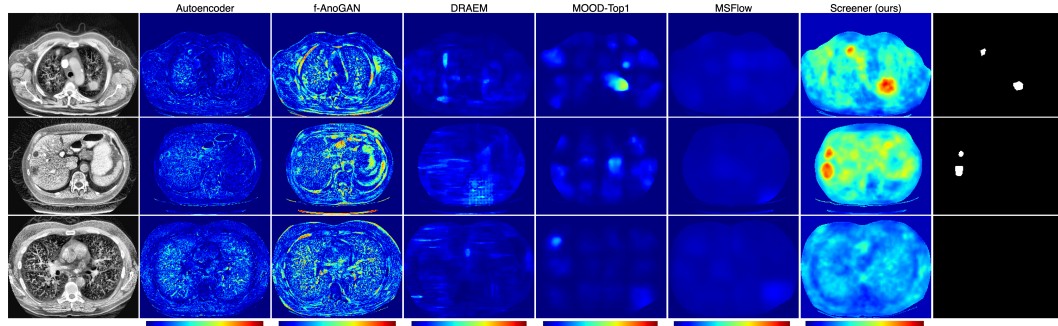

Figure 3: Qualitative comparison of baseline UVAS methods and SCREENER anomaly maps on chest and abdomen regions. First column contains CT slices, columns 2 to 6 are baseline methods, column 7 is SCREENER. Last column depicts ground trught annotation mask.

Table 3: Ablation study of the effect of conditional model for the fixed descriptor model (VICReg) and different conditional density models (gaussian and normalizing flow). None in Condtion model column means that results are given for a marginal density model.

| Descriptor model | Condition model | Density model | AUROC | | | | AUROC up to FPR0.3 | | | | AUPRO up to FPR0.3 | | | |
|---|---|---|---|---|---|---|---|---|---|---|---|---|---|---|
| | | | LIDC | MIDRC | KiTS | LiTS | LIDC | MIDRC | KiTS | LiTS | LIDC | MIDRC | KiTS | LiTS |
| VICReg, $d^{\text{desc}} = 32$ | None | Gaussian | 0.81 | 0.81 | 0.61 | 0.71 | 0.41 | 0.47 | 0.12 | 0.22 | 0.46 | 0.62 | 0.13 | 0.28 |
| | Sin-cos pos. | Gaussian | 0.82 | 0.80 | 0.74 | 0.77 | 0.45 | 0.42 | 0.26 | 0.34 | 0.40 | 0.50 | 0.27 | 0.32 |
| VICReg, $d^{\text{desc}} = 32$ | APE | Gaussian | 0.88 | 0.80 | 0.78 | 0.86 | 0.67 | 0.46 | 0.34 | 0.56 | 0.43 | 0.38 | 0.35 | 0.55 |
| VICReg, $d^{\text{desc}} = 32$ | Masking-equiv. | Gaussian | **0.96** | **0.84** | **0.87** | **0.90** | **0.90** | **0.58** | **0.58** | **0.71** | **0.64** | **0.41** | **0.57** | **0.48** |
| VICReg, $d^{\text{desc}} = 32$ | None | Norm. flow | **0.96** | **0.89** | 0.88 | 0.93 | **0.89** | **0.68** | 0.62 | 0.78 | **0.67** | **0.46** | 0.62 | 0.65 |
| VICReg, $d^{\text{desc}} = 32$ | Sin-cos pos. | Norm. flow | **0.96** | **0.89** | **0.90** | **0.94** | **0.89** | **0.68** | **0.69** | **0.80** | 0.66 | **0.46** | **0.68** | **0.66** |
| VICReg, $d^{\text{desc}} = 32$ | APE | Norm. flow | **0.96** | 0.88 | 0.89 | **0.94** | 0.87 | 0.65 | 0.67 | **0.80** | 0.64 | 0.43 | 0.66 | **0.66** |
| VICReg, $d^{\text{desc}} = 32$ | Masking-equiv. | Norm. flow | **0.96** | 0.87 | **0.90** | 0.93 | 0.88 | 0.64 | 0.68 | **0.80** | 0.65 | 0.40 | 0.67 | 0.63 |

### 4.3 MAIN RESULTS

We compare our best model (VICreg descriptor model, sin-cos positional encodings condition model and conditional normalizing flow density model) with baselines that represent different approaches to visual anomaly segmentation. Specifically, we implement 3D versions of autoencoder (Baur et al., 2021), f-anoGAN (Schlegl et al., 2019) (reconstruction-based methods), DRAEM (Zavrtanik et al., 2021), MOOD-Top1 (Marimont & Tarroni, 2023) (methods based on synthetic anomalies) and MSFlow (density-based method on top of ImageNet features). Quantitative comparison is presented in table 2. Qualitative comparison is shown in Figure 3.

The analysis of the poor performance of the reconstruction-based methods is given in Appendix B. Synthetic-based models yield many false negatives because during training they were penalized to predict zero scores in the unlabeled real pathological regions which may appear in training images. Meanwhile, MSFlow heavily relies on an ImageNet-pre-trained encoder which produces irrelevant features of 3D medical CT images. Our density-based model with domain-specific self-supervised features outperforms baselines by a large margin.

### 4.4 CONDITION AND DENSITY MODELS' ABLATION

Table 3 demonstrates ablation study of our proposed condition model. We compare our condition model with two baselines: vanïla sin-cos positional encodings and anatomical positional embeddings (Goncharov et al., 2024), described in Appendix D. We evaluate condition models in combination with different density models, described in Section 3.3. We use the VICReg descriptor with $d^{\text{desc}} = 32$ as it shows slightly better results than contrastive objective as reported in Section 4.5.

All conditioning strategies yield results similar to the unconditional model when using expressive normalizing flow density model. However, in experiments with simple gaussian density models, we see that the results significantly improve as the condition model becomes more informative.

Table 4: Ablation study of the effect of descriptor model. In these experiments we do not use conditioning and use normalizing flow as a marginal density model. We include MSFlow to demonstrate that descriptor model pre-trained on ImageNet is inappropriate for 3D medical CT images.

| Descriptor model | Condition model | Density model | AUROC | | | | AUROC up to FPR0.3 | | | | AUPRO up to FPR0.3 | | | |
|---|---|---|---|---|---|---|---|---|---|---|---|---|---|---|
| | | | LIDC | MIDRC | KiTS | LiTS | LIDC | MIDRC | KiTS | LiTS | LIDC | MIDRC | KiTS | LiTS |
| ImageNet | Sin-cos pos. | MSFlow | 0.70 | 0.66 | 0.64 | 0.64 | 0.26 | 0.20 | 0.18 | 0.17 | 0.21 | 0.14 | 0.19 | 0.17 |
| SimCLR, $d^{\text{desc}} = 32$ | None | Norm. flow | 0.96 | 0.87 | 0.87 | 0.91 | 0.90 | 0.65 | 0.58 | 0.71 | 0.68 | 0.43 | 0.58 | 0.60 |
| VICReg, $d^{\text{desc}} = 32$ | None | Norm. flow | 0.96 | 0.89 | 0.88 | 0.93 | 0.89 | 0.68 | 0.62 | 0.78 | 0.67 | 0.46 | 0.62 | 0.65 |
| VICReg, $d^{\text{desc}} = 128$ | None | Norm. flow | 0.96 | 0.90 | 0.87 | 0.93 | 0.90 | 0.72 | 0.60 | 0.77 | 0.70 | 0.52 | 0.60 | 0.65 |

Noticeably, our proposed masking-invariant condition model allows Gaussian model to compete with complex flow-based models and achieve very strong anomaly segmentation results.

### 4.5 DESCRIPTOR MODELS' ABLATION

We also ablate descriptor models in Table 4. We compare contrastive and VICReg models with $d^{\text{desc}} = 32$. To ablate the effect of the descriptors' dimensionality, we also include VICReg model with $d^{\text{desc}} = 128$. To demonstrate that our domain-specific self-supervised descriptors are better than descriptors pre-trained on general-domain we compare with MSFlow (Zhou et al., 2024).

## 5 RELATED WORK

### 5.1 VISUAL UNSUPERVISED ANOMALY LOCALIZATION

In this section, we review several key approaches, each represented among the top five methods on the localization track of the MVTec AD benchmark (Bergmann et al., 2021), developed to stir progress in visual unsupervised anomaly detection and localization.

**Synthetic anomalies** In unsupervised settings, real anomalies are typically absent or unlabeled in training images. To simulate anomalies, researchers synthetically corrupt random regions by replacing them with noise, random patterns from a special set (Yang et al., 2023), or parts of other training images (Marimont & Tarroni, 2023). A segmentation model is trained to predict binary masks of corrupted regions, providing well-calibrated anomaly scores for individual pixels. While straightforward to train, these models may overfit to synthetic anomalies and struggle with real ones.

**Reconstruction-based** Trained solely on normal images, reconstruction-based approaches (Baur et al., 2021; Kingma & Welling, 2013; Schlegl et al., 2019), poorly reconstruct anomalous regions, allowing pixel-wise or feature-wise discrepancies to serve as anomaly scores. Later generative approaches (Zavrtanik et al., 2021; Zhang et al., 2023; Wang et al.) integrate synthetic anomalies. The limitation stemming from anomaly-free train set assumption still persists—if anomalous images are present, the model may learn to reconstruct anomalies as well as normal regions, undermining the ability to detect anomalies through differences between $x$ and $\hat{x}$.

**Features pre-trained on ImageNet + density estimation** Density-based methods for anomaly detection model the distribution of the training data. Density estimation can be done in a non-parametric way by the collection of a memory bank of objects (Roth et al., 2022; Bae et al., 2023). As modeling of the distribution of raw pixel values is infeasible, these methods usually model the distribution of their deep features.

Unsupervised anomaly detection has seen the rise of flow-based methods (Serrà et al., 2019; Yu et al., 2021), which leverage normalizing flows to assign low likelihoods to anomalies. However, these methods struggle with high-dimensional raw RGB images, often assigning higher likelihoods to anomalies than normal data (Kirichenko et al., 2020). To address this, flow-based methods have been adapted to operate on high-dimensional features extracted from images. Multiscale feature processing, as seen in DifferNet (Rudolph et al., 2021) and CFlow-AD (Gudovskiy et al., 2022), enhances defect detection by handling variations in defect size. However, CFlow-AD's independent estimation of each feature vector lacks contextual awareness, resulting in fragmented and inaccurate

localization. MSFlow (Zhou et al., 2024) addresses this limitation by concurrently estimating features at all positions, incorporating contextual information through 3x3 convolutions and employing a fusion flow block for information exchange across scales.

Our method is related to FastFlow (Yu et al., 2021), CFlow (Gudovskiy et al., 2022) and MS-Flow (Zhou et al., 2024) methods for anomaly segmentation. Besides some technical differences (e.g. working with 2D natural images), there are several substantial differences: 1) these methods are based on a supervised encoder, pre-trained on ImageNet; 2) we show that density-based anomaly segmentation in medical images can be improved using data-driven condition variables.

From this family, we selected MSFlow as a representative baseline, because it is simpler than PNI, and yields similar top-5 results on the MVTec AD.

### 5.2 MEDICAL UNSUPERVISED ANOMALY LOCALIZATION

While there's no standard benchmark for pathology localization on CT images, MOOD (Zimmerer et al., 2021) offers a relevant benchmark with generated anomalies. Unfortunately, this benchmark is currently closed for submissions, preventing us from evaluating our method. We include the top-performing method from MOOD (Marimont & Tarroni, 2023) in our comparison, that relies on synthetic anomalies.

Other recognized methods for anomaly localization in medical images are reconstruction-based: variants of AE/VAE (Baur et al., 2021; Shvetsova et al., 2021), f-AnoGAN Schlegl et al. (2019), and diffusion-based (Pinaya et al., 2022). These approaches highly rely on the fact that the the training set consists of normal images only. However, it is challenging and costly to collect a large dataset of CT images of normal patients. While these methods work acceptable in the domain of 2D medical images and MRI, the capabilities of the methods have not been fully explored in a more complex CT data domain. We have adapted these methods to 3D.

## 6 CONCLUSION

This work explores fully self-supervised approach to anomaly detection and localization in medical 3D images. Previously, methods relied on supervised approaches and anomaly-free training datasets assumption, which hardly holds in typical medical scenarios. We propose SCREENER as a three component model, comprised of (i) self-supervised representation learning descriptor for image features, (ii) density-based anomaly detection model that learns distribution of the features, and (iii) conditioning model containing auxiliary information which boosts simpler density models.

Domain-specific and self-supervised SCREENER is no longer inhibited by limitations of the earlier methods and outperforms them by a large margin, which can be seen from empirical results obtained on the large-scale collection of computed tomography datasets. As our framework is modular, we learned and tested several model choices for each of the component, resulting in a comprehensive ablation study.

**Limitations**  We note that this work is largely a proof of concept for SSL in 3D medical imaging as there are still limitations to the proposed approach. Density based anomaly detection poses a limitation in that *rare* patterns can be flagged as pathological. Since rareness is highly predictive of anomaly, applying to pathology segmentation SCREENER may yield false positive errors on *healthy* but rare patterns. Another limitation concerns representativeness of the training sample. Our training dataset contains chest and abdominal CTs with much more chest samples. This causes more false positive errors in abdominal region. To work in other anatomical regions, our model needs to be trained on the corresponding images.

**Future work**  While the performance gains compared to baselines are already significant, we note that further improvements might be achieved from increasing descriptors and conditions dimensionality and experiments with multi-scale representations (e.g. by building feature pyramids). Another possible avenue for future work is to study scaling laws, i.e. self-supervised models typically scale well with increasing pretraining dataset sizes. Distillation of SCREENER into UNet at a pre-training stage is also possible and might prove effective for pathology segmentation tasks.

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

# A ANALYSIS OF RECONSTRUCTION-BASED MODELS

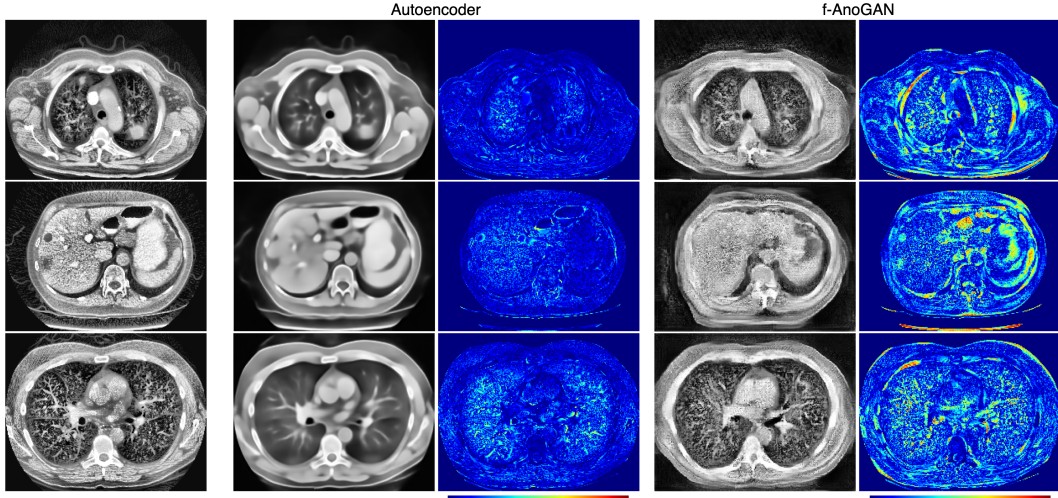

Figure 4: The figure shows 2D slices of CT images (first column) alongside reconstructions and anomaly maps generated by two methods: an Autoencoder (Baur et al., 2021) (second and third columns) and f-AnoGAN (Schlegl et al., 2019) (last two columns). Autoencoder overfits for pixel reconstruction, so it generates pathologies and fails to segment them. Also Autoencoder produces blurry generations, leading to inaccurate reconstructions of fine details and high anomaly scores on these details (e.g., vessels in the lungs). f-AnoGAN, on the other hand, avoids generating pathologies, but the generation quality still is insufficient for precise segmentation of only pathological voxels. GANs are known to be unstable and sensitive to hyperparameters, necessitating careful tuning and experimentation to achieve optimal results.

# B ANALYSIS OF RECONSTRUCTION-BASED MODELS

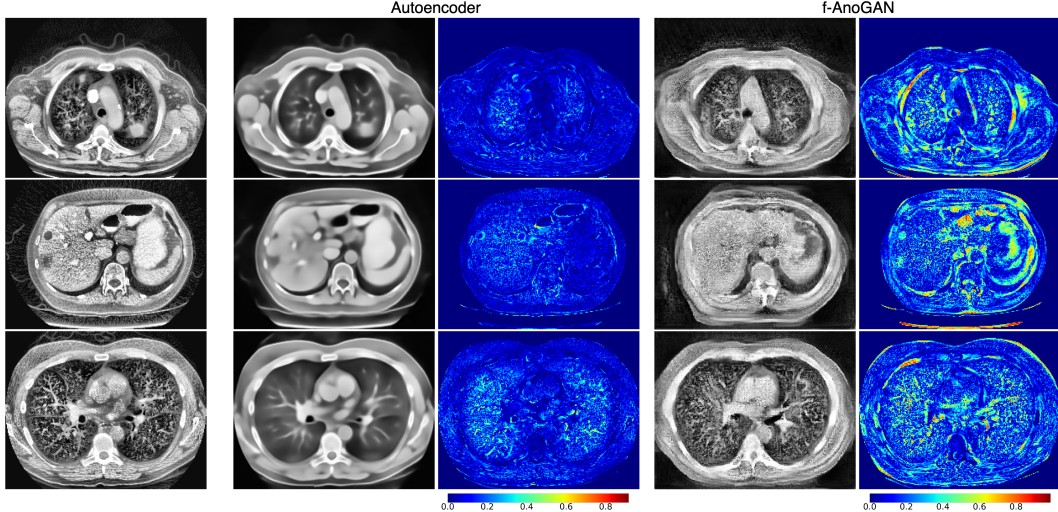

Figure 5: The figure shows 2D slices of CT images (first column) alongside reconstructions and anomaly maps generated by two methods: an Autoencoder (Baur et al., 2021) (second and third columns) and f-AnoGAN (Schlegl et al., 2019) (last two columns). Autoencoder overfits for pixel reconstruction, so it generates pathologies and fails to segment them. Also Autoencoder produces blurry generations, leading to inaccurate reconstructions of fine details and high anomaly scores on these details (e.g., vessels in the lungs). f-AnoGAN, on the other hand, avoids generating pathologies, but the generation quality still is insufficient for precise segmentation of only pathological voxels. GANs are known to be unstable and sensitive to hyperparameters, necessitating careful tuning and experimentation to achieve optimal results.

# C SELF-SUPERVISED LEARNING

Below, we outline two representative methods: SimCLR and VICReg.

**SimCLR** In contrastive models, the key objective is to maximize the similarity between embeddings of positive pairs (augmented views of the same input) while minimizing their similarity with negative pairs (views from other inputs). To this end, InfoNCE loss on embeddings is minimized:

$$\min_{\theta} \quad \sum_{i=1}^{N} \sum_{k \in \{1,2\}} - \log \frac{\exp(\langle z_i^{(1)}, z_i^{(2)} \rangle / \tau)}{\exp(\langle z_i^{(1)}, z_i^{(2)} \rangle / \tau) + \sum_{j \neq i} \sum_{l \in \{1,2\}} \exp(\langle z_i^{(k)}, z_j^{(l)} \rangle / \tau)}, \quad (1)$$

where $z_i^{(1)}$ and $z_i^{(2)}$ form a positive pair (i.e. augmentations of the same image $x_i$).

**VICReg** Non-contrastive learning avoids explicit negative pairs by structuring the embedding space directly. Specifically, VICReg objective enforces invariance among positive embeddings while constraining covariance matrix of features to be diagonal and variance to be equal to some constant:

$$\min_{\theta} \quad \alpha \cdot \mathcal{L}^{\text{inv}} + \beta \cdot \mathcal{L}^{\text{var}} + \gamma \cdot \mathcal{L}^{\text{cov}}, \quad (2)$$

The first term $\mathcal{L}^{\text{inv}} = \frac{1}{N \cdot D} \sum_{i=1}^{N} \|z_i^{(1)} - z_i^{(2)}\|^2$ penalizes embeddings to be invariant to augmentations. The second term $\mathcal{L}^{\text{var}} = \sum_{k \in \{1,2\}} \frac{1}{D} \sum_{i=1}^{D} \max\left(0, 1 - \sqrt{C_{i,i}^{(k)} + \varepsilon}\right)$ enforces individual embeddings' dimensions to have unit variance. The third term $\mathcal{L}^{\text{cov}} = \sum_{k \in \{1,2\}} \frac{1}{D} \sum_{i \neq j} \left(C_{i,j}^{(k)}\right)^2$ encourages different embedding's dimensions to be uncorrelated, increasing the total information content of the embeddings.

# D  BASELINE CONDITION MODELS

**Sin-cos positional encodings**   The existing density-based UVAS methods Gudovskiy et al. (2022); Zhou et al. (2024) for natural images use standard sin-cos positional encodings for conditioning. We also employ them as an option for condition model in our framework. However, let us clarify what we mean by sin-cos positional embeddings in CT images. Note that we never apply descriptor, condition or density models to the whole CT images due to memory constraints. Instead, at all the training stages and at the inference stage of our framework we always apply them to image crops of size $H \times W \times S$, as described in Sections 3.1, 3.3. When we say that we apply sin-cos positional embeddings condition model to an image crop, we mean that compute sin-cos encodings of absolute positions of its pixels w.r.t. to the whole CT image.

**Anatomical positional embeddings**   To implement the idea of learning the conditional distribution of image patterns at each certain anatomical region, we need a condition model producing conditions $c[p]$ that encode which anatomical region is present in the image at every position $p$. Supervised model for organs' semantic segmentation would be an ideal condition model for this purpose. However, to our best knowledge, there is no supervised models that are able to segment all organs in CT images. That is why, we decided to try the self-supervised APE Goncharov et al. (2024) model which produces continuous embeddings of anatomical position of CT image pixels.

# E  DENSITY MODELS

Below, we describe simple Gaussian density model and more expressive learnable Normalizing Flow model.

**Gaussian**   Gaussian marginal density model is written as

$$-\log q_{\theta^{\mathrm{dens}}}(y) = \frac{1}{2}(y-\mu)^{\top}\Sigma^{-1}(y-\mu) + \frac{1}{2}\log\det\Sigma + \mathrm{const}, \tag{3}$$

where the trainable parameters $\theta^{\mathrm{dens}}$ are mean vector $\mu$ and diagonal covariance matrix $\Sigma$.

Conditional gaussian density model is written as

$$-\log q_{\theta^{\mathrm{dens}}}(y \mid c) = \frac{1}{2}(y-\mu_{\theta^{\mathrm{dens}}}(c))^{\top}(\Sigma_{\theta^{\mathrm{dens}}}(c))^{-1}(y-\mu_{\theta^{\mathrm{dens}}}(c)) + \frac{1}{2}\log\det\Sigma_{\theta^{\mathrm{dens}}}(c) + \mathrm{const}, \tag{4}$$

where $\mu_{\theta^{\mathrm{dens}}}$ and $\Sigma_{\theta^{\mathrm{dens}}}$ are MLP nets which take condition $c \in \mathbb{R}^{d^{\mathrm{cond}}}$ as input and predict a conditional mean vector $\mu_{\theta^{\mathrm{dens}}}(c) \in \mathbb{R}^{d^{\mathrm{desc}}}$ and a vector of conditional variances which is used to construct the diagonal covariance matrix $\Sigma_{\theta^{\mathrm{dens}}}(c) \in \mathbb{R}^{d^{\mathrm{desc}} \times d^{\mathrm{desc}}}$.

As described in Section 3.3, at both training and inference stages, we need to obtain dense negative log-density maps. Dense prediction by MLP nets $\mu_{\theta^{\mathrm{dens}}}(c)$ and $\Sigma_{\theta^{\mathrm{dens}}}(c)$ can be implemented using convolutional layers with kernel size $1 \times 1 \times 1$. In practice, we increase this kernel size to $3 \times 3 \times 3$, which can be equivalently formulated as conditioning on locally aggregated conditions.

**Normalizing flow**   Normalizing flow model of descriptors' marginal distribution is written as:

$$-\log p_{\theta^{\mathrm{dens}}}(y) = \frac{1}{2}\|f_{\theta^{\mathrm{dens}}}(y)\|^2 - \log\left|\det\frac{\partial f_{\theta^{\mathrm{dens}}}(y)}{\partial y}\right| + \mathrm{const}, \tag{5}$$

where neural net $f_{\theta}$ must be invertible and has a tractable jacobian determinant.

Conditional normalizing flow model of descriptors' conditional distribution is given by:

$$-\log p_{\theta^{\mathrm{dens}}}(y \mid c) = \frac{1}{2}\|f_{\theta^{\mathrm{dens}}}(y,c)\|^2 - \log\left|\det\frac{\partial f_{\theta^{\mathrm{dens}}}(y,c)}{\partial y}\right| + \mathrm{const}, \tag{6}$$

where neural net $f_{\theta} : \mathbb{R}^{d^{\mathrm{desc}}} \times \mathbb{R}^{d^{\mathrm{cond}}} \to \mathbb{R}^{d^{\mathrm{desc}}}$ must be invertible w.r.t. the first argument, and the second term should be tractable.

We construct $f_\theta$ by stacking Glow layers Kingma & Dhariwal (2018): act-norms, invertible linear transforms and affine coupling layers. Note that at both training and inference stages we apply $f_\theta$ to descriptor maps $\mathbf{y} \in \mathbb{R}^{h \times w \times s \times d^{\mathrm{desc}}}$ in a pixel-wise manner to obtain dense negative log-density maps. In conditional model, we apply conditioning in affine coupling layers similar to Gudovskiy et al. (2022) and also in each act-norm layer by predicting maps of rescaling parameters based on condition maps.

