# OpenReview forum: "Screener: Learning Conditional Distribution of Dense Self-supervised Representations for Unsupervised Pathology Segmentation in 3D Medical Images"
_ICLR.cc/2025/Conference — Submitted to ICLR 2025_

### Official Review · Reviewer_5vEP · 2024-10-30

**Soundness:** 4
**Presentation:** 3
**Contribution:** 3
**Rating:** 6
**Confidence:** 5

**Summary:**

This paper introduces SCREENER, a fully self-supervised framework for anomaly detection in 3D CT images. SCREENER learns a conditional distribution of dense image patterns, leveraging a descriptor model and conditional density model to assign anomaly scores based on rarity. The method shows promising performance on several CT pathology datasets, though some aspects could benefit from further clarity, additional benchmarks, and comparisons.

**Strengths:**

Promising Results: SCREENER achieves impressive results across multiple datasets, particularly given its unsupervised nature. The method could advance anomaly detection in settings with limited labeled data.

Use of Conditional Density Modeling: The paper’s approach to conditioning on context in the density model is innovative and contributes to improving detection accuracy by simplifying anomaly scoring.

Effective Self-Supervised Learning Approach: SCREENER’s use of self-supervised pretraining on a large CT dataset provides a valuable alternative to supervised feature extractors, addressing the scarcity of annotated medical images.

**Weaknesses:**

Limited Comparative Evaluation: While SCREENER shows good performance on mixed CT datasets, it is challenging to assess its true impact due to the limited number of similar baseline methods for comparison. Including a maximum Dice score comparison to supervised SOTA or additional metrics from benchmarks like MOOD would allow for a clearer evaluation.

Lack of Standard Dataset Benchmarks: Although the authors focus on CT images, adding results on a standard brain dataset would help position SCREENER relative to established anomaly detection methods and increase its comparative strength even if this is not the primary goal of the work.

Domain Gap Influence: The effect of domain gaps between different CT datasets is not discussed. SCREENER is tested on a single combined dataset, but further analysis of domain variance (standard deviations, confidence intervals) would better inform its generalizability. Clarifying how domain gaps might impact performance is also relevant, as it remains unclear how well SCREENER would generalize to datasets with different characteristics.

Architecture and Sampling Details: The choice of downsampling size (h, w, s), upsampling, and overall architecture needs further discussion to make the design decisions transparent and reproducible.

Related Work and Citations: The paper could improve its citation of related work, including references to early synthetic anomaly detection methods (e.g., FPI) and CRADL, which has a similar structure involving a SIMCLR pretrained encoder and Gaussian/flow-based density models. Additionally, work on applied studies on anomaly scoring could enhance the background section.

Clarity and Detail in Writing: The paper is somewhat unclear, with missing details that could make its contributions and methodology more convincing. For example, some claims (such as those about density models or scale of CT data) are not fully supported, and the paper’s writing could more clearly convey the innovations and unique aspects of SCREENER.

Overlapping Test and Training Data: Some test sets, like LiTS and KiTS, are partially represented in the training dataset AbdomenAtlas, which could raise concerns about test set contamination. While this overlap may not significantly affect results, a clear discussion of this issue is necessary to address any potential impacts.

**Questions:**

Can you provide results for a maximum Dice score comparison with a supervised SOTA model to better position SCREENER’s effectiveness?

How does SCREENER perform on standard anomaly detection datasets (e.g., brain datasets), and would such benchmarks enhance the comparability of results?

Could you elaborate on the potential influence of domain gaps across different datasets in SCREENER’s performance, and is there any analysis of variance across domains?

How do choices like downsampling size, h, w, s, and upsampling affect SCREENER’s performance, and could these details be further specified?

How do you address the overlap of training and test datasets, specifically with parts of LiTS and KiTS also in AbdomenAtlas?

---

> ### Author Response · Authors · 2024-12-03
>
> Dear Reviewer **5vEP**, thank you for dedicating your time to reviewing our submission and for offering thoughtful and generally positive feedback. We are encouraged that you find our approach innovative and consider our results to be promising. We share your opinion that some additional experiments and benchmarks could be added to demonstrate the practical benefits of our method. We comment on these and other concerns below.
>
> **Dice score comparison with supervised SOTA**
>
> The key problem is that the existing CT datasets provide annotations **only for specific classes of findings**. For example, LIDC includes lung cancer masks, MIDRC provides pneumonia masks, KiTS offers kidney tumor masks, and LiTS contains liver tumor masks. However, **other pathologies present in these datasets remain unlabeled**.
>
> This limitation introduces two key issues:
> 1. **Evaluation challenges:** Standard segmentation metrics like the Dice score or object detection metrics cannot be reliably used. True positive predictions for unannotated pathologies are incorrectly counted as "false positives" when compared to the incomplete ground truth annotations. For instance, in Figure 2, our model correctly identifies pneumothorax in the second image, but because the ground truth mask does not include this finding, metrics like the Dice score are significantly penalized.
> 2. **Limited supervised models:** Supervised models can only be trained for a single annotated pathology (e.g., lung cancer). Comparing our self-supervised model, which is designed to segment all pathologies, with such supervised baselines would be inherently unfair, as they address different tasks.
>
> To address the first issue, we draw inspiration from the MVTecAD benchmark, which uses pixel-level AUROC and AUPRO metrics to evaluate unsupervised visual segmentation. These metrics are more robust to a small number of "false positive" errors, as they rely on pixel-level recall (or finding-level recall for AUPRO) and pixel-level specificity.
>
> For the second limitation, instead of directly comparing our model to supervised baselines, we propose an alternative experiment. We could distill our composite model into a standard U-Net architecture and use it as a pre-trained checkpoint. This pre-trained U-Net could then be fine-tuned for specific tasks such as lung cancer segmentation and compared against supervised baselines trained either from scratch or using other pre-trained initializations. While we were unable to include these experiments in the current submission due to time constraints, we plan to add them to the GitHub repository. Thank you for inspiring us with this idea!
>
> **Domain Gap Influence**
>
> We trained Screener on three datasets: one lung cancer screening dataset (NLST) and two abdominal datasets (AbdomenAtlas and AMOS). We then evaluated its performance on four separate datasets: two chest CT datasets (LIDC and MIDRC-RICORD-1a) and two abdominal datasets (KiTS and LiTS). While there are noticeable domain gaps between the training and testing datasets, as well as among the testing datasets themselves, the results in Table 2 demonstrate strong generalization. Screener achieves pixel-level AUROCs of approximately **0.9–0.95 on each of four testing datasets**, indicating its robustness in handling diverse chest and abdominal CT data.
>
> **Overlapping Test and Training Data**
>
> We argue that our training and test datasets are entirely distinct, with no overlap. In particular, in the paper about the AbdomenAtlas dataset, authors contrast it to the pre-existing KiTS and LiTS datasets. Therefore, we assume that there is no intersection between AbdomenAtlas and either the KiTS or LiTS datasets.
>
> **Architecture and Sampling Details**
>
> All the implementation details, e.g. neural network architectures, preprocessing, hyperparameters will be available in the github repository.
>
> **Benchmarking on brain MRI datasets**
>
> We acknowledge that benchmarking our model on brain MRI datasets is desirable. Unfortunately, we cannot include it in the current version, due to the time limitations.

---

### Official Review · Reviewer_px7H · 2024-11-03

**Soundness:** 3
**Presentation:** 2
**Contribution:** 2
**Rating:** 5
**Confidence:** 3

**Summary:**

The paper presents SCREENER, a self-supervised framework for anomaly segmentation in 3D medical CT images, specifically aimed at pathology detection. SCREENER learns conditional distributions of dense self-supervised representations, assigning higher anomaly scores to patterns with low conditional probability. This method includes three components: (1) a descriptor model that encodes local patterns, (2) a condition model encoding global context, and (3) a density model estimating the likelihood of descriptors, yielding anomaly scores for segmentation. Trained on 30,000 CT scans, SCREENER outperformed existing unsupervised anomaly segmentation methods across multiple pathologies, showcasing its potential in large-scale, label-scarce medical imaging tasks.

**Strengths:**

+ **Comprehensive Introduction and Related Work:** The introduction and related work sections are thorough and well-written. They offer a solid overview of existing methods in medical anomaly segmentation and clearly highlight their limitations, supported by relevant examples from the literature.

+ **Extensive Training Dataset:** The model is trained on a dataset of over 30,000 3D CT scans, which is a significant advantage for self-supervised learning. This large dataset aids in accurately modeling the distribution of healthy CT images, allowing the model to distinguish normal from abnormal patterns more effectively.

**Weaknesses:**

- **Unclear Illustration of the Proposed Method:** The method section lacks clarity on what parts are novel contributions versus prior work. For example, Section 2.3 largely describes existing techniques, such as SimCLR and VICReg, but it’s unclear how much these influence the new SCREENER model. The paper would benefit from clearly marking new contributions at the start of each subsection and relegating baseline methods to the experimental settings section. A figure illustrating the architecture would greatly improve comprehensibility.

- **Inappropriate Metric Selection:** The paper focuses on anomaly segmentation, yet uses anomaly detection metrics (AUROC) rather than segmentation-specific metrics, such as Dice Similarity Coefficient (DSC), Intersection over Union (IoU), Hausdorff Distance (HD), or Normalized Surface Distance (NSD). Without these, it’s difficult to assess the model's effectiveness in precise segmentation of anomalies.

**Questions:**

1. Clarify Novelty in SCREENER: The proposed SCREENER method claims to modify the density-based UVAS framework but lacks clarity on what modifications were made and their specific advantages. Clear differentiation of new versus existing components would help readers understand the proposed method’s originality.

2. Dense Features Learned by SimCLR: The paper states that SimCLR learns “dense features” (lines 81-82), but this is misleading as SimCLR typically captures global features to distinguish one image from another. Defining what is meant by "dense features" here and how SimCLR is adapted for this purpose would improve accuracy.

3. Define Key Terms: Essential terms like "conditional distribution," "local image patterns," and "global image content" are insufficiently explained, making the method challenging to follow. Definitions and possibly examples would clarify the underlying assumptions and techniques used.

4. Figure 1 Anomaly Map Clarity: In Figure 1, the second column shows an image flagged as diseased (as indicated by attention maps), yet lacks a mask. A mask in this column or clearer labeling would clarify the model's detection process.

5. AUROC Use for Segmentation Tasks: AUROC is not ideal for segmentation tasks, which typically require spatial accuracy. Metrics like HD or NSD would provide more nuanced insights into segmentation quality, particularly for edge or boundary delineation.

---

> ### Author Response · Authors · 2024-11-30
>
> Dear Reviewer **px7H**, thank you for reviewing our submission and for offering thoughtful feedback. We greatly value your insights and have addressed your concerns below.
>
> **Clarifying novelty & improving method presentation**
>
> You correctly pointed out that our novel contributions were not clearly distinguished from prior work. Additionally, our citation of SimCLR in the context of dense SSL was misleading, and terms “local patterns” and “global context” were not well explained.
>
> To address this, we have made the following revisions:
>
> 1. Added a Background & notation section (Section 2) to outline the existing density-based UVAS framework, involving supervised descriptor model, density model, and conditioning on vanïla sin-cos positional encodings.
> 2. In the Background & notation section we also discuss works on self-supervised learning (e.g., SimCLR and VICReg), and dense self-supervised learning (e.g., VADER, DenseCL, VICRegL), which are closely related to our method. We also replace misleading citations in lines 81-82 with correct citations of dense SSL methods.
> 3. In Method section (Section 3), we emphasize our two key contributions: *self-supervised* descriptor model and *condition model*. Suitability of self-supervised features within the density-based UVAS framework is a non-trivial finding which contributes to the field. The concept of a condition model which learns data-driven variables for conditioning is also novel within the context of UVAS literature.
> 4. Added Figure 2, illustrating the main components of our method: descriptor model, condition model and density model.
> 5. In Section 3.1 we provide intuition and details of the pre-training procedure for our self-supervised descriptor model. It produces dense feature maps that capture all the mutual information between different augmented crops preserving “local” image content. For example, the lung cancer mass visible in both crops in the upper part of Figure 2. Therefore, descriptor model’s feature maps contain information about the presence of pathologies.
> 6. In Section 3.2 we describe our condition model. Similarly to the descriptor model, it produces dense feature maps capturing all the mutual information between different augmented *masked* crops. We call this information “global”, because it can be inferred from masked image views, which miss the “local” image content. For example, note that the lung cancer mass is not visible in the first masked crop in the middle part of Figure 2. Thus, information about the presence of pathologies is excluded from the condition model’s feature maps.
> 7. In Section 3.3 we describe the details of our density model, which can be viewed as the predictive model, predicting the descriptor model’s feature maps from the condition model’s feature maps.
>
> **Metrics**
>
> Your concerns regarding our evaluation metrics are completely valid and understandable. We acknowledge that pixel-level AUROC is not the most representative metric to assess the segmentation quality. Ideally, we would prefer to evaluate our model using segmentation metrics such as Dice score or Hausdorff distance, object detection metrics like finding-level Average Precision, and classification metrics such as patient-level AUROC.
>
> However, it is challenging to fairly estimate these metrics due to the limitations of the available test datasets. These datasets provide annotations **only for specific classes of findings** — for instance, LIDC includes only lung cancer masks, MIDRC contains only pneumonia masks, KiTS provides kidney tumor masks, and LiTS offers liver tumor masks — while **other pathologies present in these datasets remain unlabeled**. As a result, our model’s true positive predictions for unannotated pathologies are inadvertently counted as “false positives” when compared with the available partial annotations. For example, in Figure 2, our model correctly predicts pneumothorax in the second image, but the ground truth mask does not include this finding, which can significantly reduce metrics such as the Dice score.
>
> To address this limitation, we draw inspiration from the MVTecAD benchmark, which uses pixel-level AUROC and AUPRO metrics to assess the quality of unsupervised visual segmentation. These metrics are less sensitive to a small number of “false positive” errors because they depend on the pixel-level recall (or finding-level recall in the case of AUPRO) and pixel-level specificity. We calculate recall with respect to only the annotated pathologies. We estimate specificity using a random subsample of voxels that do not belong to the annotated pathologies. This yields quite a tight lower bound since most of these voxels are indeed normal.
>
> **Improving Figure 1**
>
> Following your suggestion, we updated Figure 1 to highlight that our model correctly identifies pneumothorax in the second image. The ground truth mask misses this pathology, as the image originates from the LIDC dataset, which provides only lung cancer annotations.

---

### Official Review · Reviewer_jnQD · 2024-11-03

**Soundness:** 2
**Presentation:** 1
**Contribution:** 2
**Rating:** 5
**Confidence:** 3

**Summary:**

This paper introduces a self-supervised method for anomaly segmentation in 3D medical images. The model architecture includes a descriptor, condition, and density module. However, the paper lacks clarity in its presentation, with limited explanations and no figures to illustrate the pipeline, making it challenging to grasp the overall workflow and novelty of the approach.

**Strengths:**

1. This work introduces a large-scale dataset on pathology segmentation, integrating data from NLST, AMOS. AbdomenAtlas.
2. This work aims to solve the meaningful problem of self-supervised pathology segmentation.
3. Lacks the statistics of the claimed large dataset. The author should summarize and describe this dataset in detail.

**Weaknesses:**

1. This work is poorly written and lacks clarity in conveying its high-level concept. The descriptions of the descriptor, condition, and density models are unclear, and there are no experiments demonstrating the effectiveness of these components.
2. A pipeline illustration figure is recommended to show the workflow of this work.

**Questions:**

1. Add a figure to show the pipeline of this work.
2. Add the summary for the dataset statistics, such as size, density, pathology distribution.
3. Add experiments/qualitative comparison to demonstrate the effectiveness of proposed components

---

> ### Author Response · Authors · 2024-11-23
>
> Dear Reviewer **jnQD**, thank you for taking the time to review our submission and for providing valuable feedback. We appreciate your insights and have addressed your concerns below.
>
> **Clarity and Illustration**
>
> We acknowledge that our initial submission may not have fully conveyed the high-level concept of our method. To enhance clarity, we have added **Figure 2** illustrating the overall workflow of our approach, as you recommended. The high-level concept is further elaborated in Section 2.1, where we outline how our method builds upon the existing density-based framework for unsupervised visual anomaly segmentation (UVAS).
>
> Novelty of our method is two-fold:
> 1. **Self-Supervised Descriptor model:** Existing density-based UVAS methods employ supervised feature extractors, which we replace with a self-supervised one. We demonstrate that density estimation in a self-supervised feature space is an effective approach to anomaly segmentation — a non-trivial finding that contributes to the field.
> 2. **Self-Supervised Condition Model:** Existing density-based UVAS methods are either unconditional or use very simple conditioning on standard sin-cos positional encodings. To estimate density of visual features they need to employ highly expressive Normalizing Flows.  We introduce a trainable self-supervised condition model that learns masking-invariant feature maps. Conditioning on these data-driven conditions allows us to achieve SotA results using a simple Gaussian density model.
>
> Detailed descriptions of the descriptor, condition, and density models are provided in Sections 2.2, 2.3, and 2.4, respectively. We believe these sections offer comprehensive explanations of our method's components. If there are specific aspects that remain unclear, we would greatly appreciate further guidance on how we can clarify them.
>
> **Experiments demonstrating effectiveness of the components**
>
> To demonstrate the effectiveness of the proposed components, we have conducted extensive experiments for component ablation study:
> - **Descriptor Model Ablation:** **Table 4** presents an ablation study of the descriptor model, including a comparison with MSFlow — a state-of-the-art density-based UVAS method that employs a supervised descriptor model.
> - **Condition and Density Models Ablation:** **Table 3** provides an ablation study of the condition and density models.
>
> These experiments highlight the contributions of each component to the overall performance of our method. We hope these results address your concerns regarding the experimental validation of our approach. If there are additional experiments or comparisons you believe would strengthen our work, we welcome your suggestions.
>
> **Dataset Statistics**
>
> We would like to clarify that we do not introduce a new dataset in our work. Instead, we train our model on a collection of publicly available datasets, specifically NLST, AMOS, and AbdomenAtlas. We have updated **Table 1** to include the numbers of images with non-zero pathology masks.
>
> Regarding pathology distribution, calculating statistics is challenging due to the following reasons:
> - The training datasets **do not contain annotations of pathologies** — neither labels nor masks (that's why self-supervised learning is employed).
> - Each test dataset provides annotations for **only a specific class** of pathologies (e.g., LIDC contains only lung cancer masks, MIDRC contains only pneumonia masks, KiTS contains only kidney tumor masks, and LiTS contains only liver tumor masks).
>
> While we acknowledge that large, uncurated CT datasets inherently include cases with various pathologies, detailed statistics on pathology distribution are not feasible without comprehensive annotations. For example, in Figure 1, the second image from the left displays a pneumothorax (the completely black region in the lung), which is not annotated in the ground truth mask because the LIDC dataset provides annotations only for lung cancer nodules.
>
> Thank you again for your thoughtful review. We believe that these revisions have significantly improved the clarity and completeness of our manuscript. If there are specific areas where you feel further improvements can be made, we would be grateful for your additional feedback.

---

> ### Comment · Reviewer_jnQD · 2024-11-26
>
> Thank you for your response. I appreciate that Figure 2 has been added to illustrate the pipeline of this work. However, some of my concerns remain only partially addressed:
>
> As shown in Figure 2, the training process for both the depictor model and the conditional model appears to follow standard procedures, incorporating two augmentation methods respectively. How do the authors ensure that the depictor model extracts informative feature maps that are invariant to image crops and color variations? Additionally, the feature maps on the right side of Figure 2 for both the depictor model and the conditional model seem identical, which may require further clarification or revision.
>
> Nonetheless, I have slightly raised my score from 3 to 5.

---

> ### Author Response · Authors · 2024-11-26
>
> Dear Reviewer **jnQD**,
>
> As described in Sections 2.2 and 2.3, we train both descriptor and condition model to produce feature maps (maps of pixel-level embeddings) using dense contrastive (or VICReg) objective. These objectives enforce embeddings in positive pairs to be similar and embeddings in negative pairs to be apart from each other.
>
> We define a *positive pair* as any pair of pixel-level embeddings obtained from different augmented crops but having the same absolute position w.r.t. the seed image (we depict such embeddings by the same color in Figure 2). Thus, pooling together embeddings in positive pairs ensures that feature maps are equivariant to image crops and invariant to color variations.
>
> We call a pair of pixel-level embeddings *negative* if they have different absolute position w.r.t. to the seed image or if they are obtained from different seed images. Pushing apart embeddings in negative pairs helps to avoid representation collapse (constant non-informative feature maps), thus making feature maps informative.
>
> The only difference between our descriptor and condition model is the augmentations that we use for positive pairs generation.
>
> When generating positive pairs for the condition model, we use random masking of image blocks in addition to crops and color jitter. Thus, condition model learns *masking-invariant* informative feature maps. This means that these feature maps are predictable from all masked image views. However, presence of anomalies is not predictable from all masked image views (for example lung mass is not visible in the first masked augmented crop in Figure 2). Therefore, masking-invariant feature maps do not contain information about presence of anomalies.
>
> On the contrary, when training the descriptor model, we ensure that all the augmentations preserve local image semantics. For example, lung mass is visible in both unmasked augmented crops in Figure 2. Thus, descriptor model's feature maps are allowed to distinguish between lung mass and normal lung.
>
> As a result, you can see in Figure 2 that feature maps produced by descriptor model are more fine-grained, while condition model's feature maps are much smoother.
>
> As we mention in Section 2.1, density model can be viewed as a predictive model that tries to predict descriptor model's feature maps from condition model's feature maps. High anomaly scores can be interpreted as high prediction errors. Our empirical results show that
> 1) Descriptor model indeed produces different features in normal and abnormal image regions (otherwise, density model would not assign different anomaly scores to them).
> 2) Condition model does not contain information about presence of pathologies (otherwise, density model would yield low prediction errors in pathological regions)
> 3) Our self-supervised condition model contains more information than baseline condition models (standard positional encodings, APE), which simplifies conditional distributions of descriptor model's pixel-level embeddings given condition model's pixel-level embeddings, allowing to use simply gaussian density model.

---

> > ### Author Response · Authors · 2024-12-01
> >
> > Dear reviewer **jnQD**,
> >
> > To improve our method presentation we have further revised our manuscript as follows:
> >
> > 1. Added a Background & notation section (Section 2) to outline the existing density-based UVAS framework, involving supervised descriptor model, density model, and conditioning on vanïla sin-cos positional encodings.
> > 2. In the Background & notation section we also discuss works on self-supervised learning (e.g., SimCLR and VICReg), and dense self-supervised learning (e.g., VADER, DenseCL, VICRegL), which are closely related to our method. We also replace misleading citations in lines 81-82 with correct citations of dense SSL methods.
> > 3. In Method section (Section 3), we emphasize our two key contributions: self-supervised descriptor model and condition model. Suitability of self-supervised features within the density-based UVAS framework is a non-trivial finding which contributes to the field. The concept of a condition model which learns data-driven variables for conditioning is also novel within the context of UVAS literature.
> > 4. In Section 3.1 we provide intuition and details of the pre-training procedure for our self-supervised descriptor model. It produces dense feature maps that capture all the mutual information between different augmented crops preserving “local” image content. For example, the lung cancer mass visible in both crops in the upper part of Figure 2. Therefore, descriptor model’s feature maps contain information about the presence of pathologies.
> > 5. In Section 3.2 we describe our condition model. Similarly to the descriptor model, it produces dense feature maps capturing all the mutual information between different augmented masked crops. We call this information “global”, because it can be inferred from masked image views, which miss the “local” image content. For example, note that the lung cancer mass is not visible in the first masked crop in the middle part of Figure 2. Thus, information about the presence of pathologies is excluded from the condition model’s feature maps.
> > 6. In Section 3.3 we describe the details of our density model, which can be viewed as the predictive model, predicting the descriptor model’s feature maps from the condition model’s feature maps.
> >
> > We hope that these revisions address your concerns regarding the novelty of our method and clarity of the presentation. Please let us know if you consider raising your score.

---

### Author Response · Authors · 2024-12-04

Dear Reviewers **jnQD**, **px7H**, and **5vEP**,

We sincerely appreciate the time and effort you have invested in reviewing our submission.

We are encouraged that you have recognized the main **contributions** of our work:

- **Fully Self-Supervised Framework for visual anomaly detection (Reviewer 5vEP)**: We show that dense self-supervised representations are favorable alternative to supervised feature extractors in the density-based framework for unsupervised visual anomaly segmentation (UVAS). As a result, the proposed framework is fully self-supervised and applicable in the domains where labeled data is limited.

- **Innovative conditioning on self-supervised visual features (Reviewer 5vEP)**: We further extend the density-based UVAS framework by showing that instead of hand-crafted conditioning, e.g. on positional encodings, one can learn data-driven self-supervised condition variables. Conditioning on these representations simplifies true conditional distribution and allows us to achieve remarkable anomaly segmentation results with a very simple gaussian model of conditional density.

- **Large-scale empirical study and strong results (Reviewers jnQD, px7H, 5vEP)**: This paper presents the first large-scale study of UVAS methods in 3D medical CT images. We show that the proposed density-based framework outperforms other UVAS methods on unsupervised semantic segmentation of a wide range of pathologies in different anatomical regions, demonstrating strong generalization capabilities.

Your insightful feedback has been invaluable in refining our work, and we are grateful for the opportunity to address your concerns. Below, we summarize how we have incorporated your suggestions to enhance the clarity and overall quality of our manuscript.

**Clarity and Presentation Enhancements**

- **Added Method Illustration (Reviewers jnQD and px7H):** We have included **Figure 2** to visually depict the overall workflow of SCREENER, clearly illustrating the interactions between the descriptor model, condition model, and density model.
- **Added “Background & notation” Section (Reviewers jnQD, px7H, 5vEP):** A new “Background & notation” section (Section 2) has been added to outline existing frameworks (density-based UVAS and dense SSL) and position our innovations within the context of the literature.
- **Revised “Method” Section (Reviewers jnQD, px7H, 5vEP):** We have restructured the “Method” section (Section 3) to distinctly highlight our novel contributions versus prior work. Sections 3.1 to 3.3 now offer in-depth explanations of the descriptor model, condition model, and density model, respectively.
- **Improved Figure 1 (Reviewer px7H):** We have clarified that the second image in Figure 1 contains pneumothorax which is not annotated in the ground truth, but is correctly identified by our model.
- **Updated Dataset Information (Reviewer jnQD):** Table 1 has been updated to include detailed statistics of the datasets used, specifying the number of images with non-zero pathology masks.

**Choice of Evaluation Metrics (Reviewers px7H and 5vEP)**

We acknowledge the limitations of using pixel-level AUROC for segmentation tasks. However, due to partial and class-specific annotations in available datasets, metrics like Dice score are challenging to compute fairly. That's why, we followed the evaluation protocol from the MVTecAD benchmark, utilizing pixel-level AUROC and AUPRO metrics.

**Demonstrating the Effectiveness of Components (Reviewer jnQD)**

We have conducted extensive experiments, including ablation studies presented in Tables 3 and 4, to validate the effectiveness of each component of SCREENER. These studies demonstrate the contributions of the descriptor model, condition model, and density model to the overall performance.

**Addressing Domain Gap and Generalization (Reviewer 5vEP)**

- **Performance Across Different Datasets:** Despite inherent domain gaps between the training and test datasets, SCREENER achieves high performance on diverse chest and abdominal CT datasets.
- **No Overlap Between Training and Test Sets:** We verified that there is no overlap between our training datasets (NLST, AMOS, AbdomenAtlas) and test datasets (LIDC, MIDRC-RICORD-1a, KiTS, LiTS). This ensures the validity of our evaluation and addresses concerns about potential test set contamination.

**Concluding Remarks**

We believe that these revisions have substantially strengthened our manuscript by addressing the key concerns raised during the review process. We are confident that SCREENER represents a significant advancement in unsupervised visual anomaly segmentation and holds considerable promise for real-world medical imaging applications.

Thank you once again for your valuable insights and constructive suggestions. We are grateful for your consideration and hope that our revisions meet your expectations.

---

### Meta-Review · Area_Chair_Ctzn · 2024-12-16

**Metareview:**

This paper introduces a fully self-supervised framework, dubbed Screener, for 3D medical image anomaly segmentation. In this work, self-supervised representation is employed to completely eliminate the need for manual labeling, reducing this labor-intensive requirement for large-scale datasets. Experimental analysis shows promising results in fully self-supervised medical image anomaly segmentation.

This paper received 1x marginally above the acceptance threshold and 2x marginally below the acceptance threshold from reviewers. The questions raised by reviewers regarding this work centered around the clarity of the research question, methodology design, and key term definition. Although the authors addressed most of the concerns during the rebuttal phase, one key problem, i.e., the evaluation metrics, has not achieved consensus between the reviewer and the authors. As reviewer px7H suggested, several publicly available datasets provide per-voxel annotations for lesions across multiple organs and thus this inconsistency should be further considered and addressed. Meanwhile, I notice that the methods used for comparison are somehow limited and more recent SOTA methods are suggested to be included.

Therefore, although the clarity and readability have been largely improved during the discussion phase, further improvements are needed to enhance the comprehensiveness of this work. Rejection is recommended.

**Additional Comments On Reviewer Discussion:**

During the discussion, several reviewers suggested further improvement of the text and figure for better readability and formulation of the research question. The authors have thoroughly addressed these concerns by adding figures and additional descriptions, which has enhanced the quality of this work. However, major concerns regarding experiments are not fully addressed.

---

### Decision · Program_Chairs · 2025-01-22

Reject